# Bone Turnover Marker for the Evaluation of Skeletal Remodelling in Autosomal Recessive Osteopetrosis after Haematopoietic Stem Cell Transplantation: A Case Report

**DOI:** 10.3390/children10040675

**Published:** 2023-04-03

**Authors:** Máté Horváth, Orsolya Horváth, Csaba Kassa, Gabriella Kertész, Vera Goda, Lidia Hau, Anita Stréhn, Krisztián Kállay, Gergely Kriván

**Affiliations:** 1Károly Rácz Doctoral School of Clinical Medicine, Semmelweis University, Budapest, Üllői út 26, H-1085 Budapest, Hungary; 2Pediatric Haematology and Stem Cell Transplantation Unit, Central Hospital of Southern Pest National Institute of Haematology and Infectious Diseases, Albert Flórián Street 5-7, H-1097 Budapest, Hungary; orsolyahorvath.mail@gmail.com (O.H.); kassa.csaba@gmail.com (C.K.); kerteszgabi77@gmail.com (G.K.); veragoda70@gmail.com (V.G.); haulidi@gmail.com (L.H.); strehnanita@gmail.com (A.S.); dr.kallay@gmail.com (K.K.); krivang@hu.inter.net (G.K.)

**Keywords:** autosomal recessive osteopetrosis, metabolic bone disease, haematopoietic stem cell transplantation, bone turnover marker, osteoclast, C-terminal telopeptide (CTX)

## Abstract

**Background**: Autosomal recessive osteopetrosis (ARO) is a rare genetic disorder of bone metabolism, primarily affecting the remodelling function of osteoclasts. Haematopoietic stem cell transplantation (HSCT) is the first-line treatment for ARO. Traditional tools for the assessment of therapeutic response, such as measuring donor chimerism, do not provide information on bone remodelling. The use of bone turnover markers (BTMs) might be ideal. Here, we report a case of a paediatric ARO patient undergoing successful HSCT. **Methods:** For the evaluation of donor-derived osteoclast activity and skeletal remodelling throughout the transplantation, the bone resorption marker β-CTX (β-C-terminal telopeptide) was used. **Results:** The low baseline level of β-CTX markedly increased after transplantation and remained in the elevated range even after 3 months. Donor-derived osteoclast activity reached its new baseline level around the 50th percentile range after 5 months and proved to be stable during the 15-month follow-up time. The apparent increase of the baseline osteoclast activity after HSCT was in consonance with the radiographic improvement of the disease phenotype and the correction of bone metabolic parameters. Despite the successful donor-derived osteoclast recovery, craniosynostosis developed, and reconstructive surgery had to be performed. **Conclusions:** The use of β-CTX may be of aid in assessing osteoclast activity throughout the transplantation. Further studies could help to establish the extended BTM profile of ARO patients using the available osteoclast- and osteoblast-specific markers.

## 1. Introduction

Several rare genetic disorders are referred to as osteopetrosis, which have the common ground of skeletal sclerosis and abnormal osteoclast function [1]. The impairment or the complete lack of bone matrix resorption leads to abnormal bone density and skeletal malformations. Different inheritance patterns, a variety of genetic defects, and clinical presentations describe a quite heterogeneous disease group. Autosomal recessive osteopetrosis (ARO), i.e., less-frequently termed malignant infantile osteopetrosis, is a severe form of skeletal dysplasia. Characteristic clinical features of ARO include neurologic symptoms, progressive vision loss due to optic nerve compression, bone marrow failure, pathologic fractures, growth retardation, and abnormal skeletal configuration [1]. Symptoms start to develop during the first few months of life, but neonatal onset is not uncommon either. Before the era of haematopoietic stem cell transplantation (HSCT), long-term survival was rarely achieved. Frequent infections, predominantly osteomyelitis after orthopaedic surgeries, dependence on blood transfusions, and progressive neurodegeneration often led to poor quality of life or early death.

Since osteoclasts are derived from granulocyte-monocyte progenitor cells [2], HSCT has the possibility of rescuing the disease phenotype related to abnormal bone turnover [3]. The success of this choice of treatment has been improving since the early 2000s. A survival rate of 93% with T-cell replete matched donor and 80% with haploidentical donor was observed in European transplantation centres in 2015 [3]. Based on these results, HSCT has become the first-line treatment for severe ARO.

Using bone turnover markers (BTMs) for the evaluation of therapeutic response and success during HSCT is an unexplored possibility. BTMs are secreted molecules or by-products of bone formation and resorption [4]. Monitoring of osteoblast and osteoclast activity becomes available by assessing these biochemical markers from serum or urine. BTMs have well-reviewed and growing literature in primary and secondary bone metabolism disorders, especially in osteoporosis [5]. Currently, the BTM profile of ARO patients undergoing HSCT remains unknown.

In our case report, the BTM C-terminal telopeptide of type I collagen (CTX) was used to monitor bone remodelling throughout the transplantation. CTX is located on the carboxy-terminal telopeptide region of the most abundant bone-forming protein [6]. The telopeptide is liberated into circulation during the process of bone resorption mediated by cathepsin K of osteoclasts. Telopeptide concentrations correlate with the resorption activity and can be measured from both serum and urine. After its formation, CTX undergoes isomerisation, and over time, the original α isomerised collagen degradation by-product converts to β isomerised form (β-CTX).

## 2. Case Presentation

### 2.1. Initial Treatment

A six-month-old male patient diagnosed with ARO was admitted to our transplantation unit. The patient’s clinical presentation was characteristic of the disease: deformed sclerotic skull, multiple fractures, optic nerve atrophy with progressive visual impairment, and thrombocytopaenia with hepatosplenomegaly as the sign of bone marrow involvement. The auditory system is often affected in ARO, but the patient showed no sign for otologic involvement of the disease at this time. Diagnostic radiography of the chest and cranial MRI can be seen in Figure 1.

Sanger sequencing revealed the pathognomonic mutation of the osteoclast-specific gene TCIRG1, which accounts for approximately half of the ARO cases with identified genetic origin [7]. The patient was heterozygous for two variants in TCIRG1: c.1384_1386del, p.(Asn462del), which is classified as pathogenic, and c.504-6C>A, which is a variant of uncertain significance.

Due to the severity and fast progression of the disease, urgent HSCT was decided. The patient’s transplantation was performed following the guidelines of the ESID (European Society for Immunodeficiencies) and EBMT (European Society for Blood and Marrow Transplantation) working party of inborn errors [3]. The myeloablative conditioning regimen consisted of busulfan (between days –8 and –5, total dose 60 mg), thiotepa (10 mg/kg on day –5), fludarabine (40 mg/m^2^ once daily between days –6 and –3), and anti-thymocyte globulin (between days –8 and –6, total dose 10 mg/kg). Upon completing the conditioning protocol, peripheral stem cell graft was administered from an HLA (human leukocyte antigen)-matched unrelated donor. An additional stem cell boost was given 30 days after the transplantation. An absolute neutrophil cell count of >500/μL was reached on day 20, and significant improvement of the haematopoietic recovery was observed after the stem cell boost. Transplantation-associated complications characteristic of ARO, such as rejection, veno-occlusive disease, pulmonary hypertension, and severe hypercalcaemia, were not detected. HSCT-related toxicity was well tolerated. Chimerism analyses showed that >95% of peripheral blood cells were derived from the donor.

### 2.2. Follow-Up

Resolution of osteopetrosis could be seen on MR imaging that took place 6 months after the transplantation, and radiography also demonstrated the recovery of the disease phenotype (Figure 2A).

The vision loss consequent to optic nerve compression was deemed irreversible, but the auditory system was unaffected based on brainstem auditory evoked potential (BAEP) analysis and tympanometry. The patient’s neuromotor development was not delayed compared to his age group.

However, radiological normalisation was not apparent in one region, the calvaria: neurocranial sutures progressed to ossify even though osteoclast resorptive function had been initiated in other parts of the skull and the skeletal system (Figure 2B). Neurosurgery was confronted with the necessity of craniotomy considering the danger of increased intracranial pressure secondary to craniosynostosis. The reconstructive operation was successfully performed. In the 15th month, at the time of the writing of the manuscript, the patient remained on an adequate course of neuromotor development, had stable haematopoiesis, and achieved recovery of the disease phenotype related to bone metabolism.

### 2.3. β-CTX and Bone Metabolic Profile during HSCT

β-CTX measurements were made from serum samples using β-CrossLaps^®^ electro-chemiluminescence immunoassay (Beta-CrossLaps Roche Elecsys, 11972308122, Roche Diagnostics). The sample collection had started before the conditioning protocol was initiated and continued through the early and late post-transplantation period. β-CTX serum concentration curves are presented in Figure 3.

Paediatric reference intervals of BTMs were used for the evaluation [8]. Basic parameters of bone metabolism were also evaluated (calcium, phosphate, alkaline phosphatase-ALP, bone-specific alkaline phosphatase–BAP, and osteocalcin-OC) within our clinical laboratory (Table 1).

Before conditioning, baseline β-CTX concentration was below the 3rd percentile, indicating a low level of osteoclast bone resorptive activity. From day 28, a progressive increase could be interpreted: the β-CTX concentration markedly increased compared to the baseline and rose above the 97th percentile. On day 30, an additional stem cell boost was given. Osteoclast activity and β-CTX levels continued to remain in a highly elevated range even after 50 days of transplantation, and the continuously improving haematopoiesis demonstrated the successful remodelling of the bone marrow. After 2 months, β-CTX levels started to gradually decrease below the 97th percentile range. Even after 100 days, at the time of reaching the cell count of CD4+ T-lymphocytes 200/μL, β-CTX levels remained elevated. Donor-derived osteoclast activity reached its steady state after 5 months. Around this time, the conclusion of radiologic improvement could be made based on skull radiography and MR imaging. β-CTX levels remained stable around the 50th percentile margin from that onward. The influence of donor-derived osteoclasts on the β-CTX kinetics was demonstrated by the patient’s chimerism analysis, which indicated 100% donor presence throughout the follow-up period.

Pre-transplantation metabolic parameters were shown to be characteristic of the laboratory profile of ARO: minor hypocalcaemia and hypophosphatemia throughout the aplastic period with increased levels of ALP and BAP. The total calcium level increased to the physiologic range parallel to the β-CTX increment, but abnormally high calcium levels were not observed. Hypophosphatemia was also gradually corrected. Both ALP and BAP remained elevated in the first weeks but normalised gradually within three months. The osteocalcin level was within normal range throughout the course of HSCT apart from a transitory decrease in the first month.

## 3. Discussion

The process of skeletal remodelling and changes in bone turnover induced by HSCT is not well documented [9]. Regarding the post-transplantation period of ARO patients, there is a limited number of reports available [10,11,12,13]. According to these, the resolution of the pathological radio morphology and normalisation of bone density a few months after HSCT can be observed. A more recent retrospective study using qualitative X-ray assessment and biochemical laboratory markers demonstrated that functional normalisation of osteoclasts occurs by the time of haematopoietic recovery [12]. Improved bone turnover results in significant changes in mineral density and bone morphology, although the radiological resolution is not quite complete in the early months of stem cell transplantation. The published literature provides scant information on the long-lasting effects of this bone remodelling and the long-term follow-up.

The use of BTMs as diagnostic or therapy-monitoring tools in metabolic bone disorders has been receiving more attention in recent years [4,5,6]. The bone resorption marker β-CTX was used for the assessment of bone remodelling in the case presented, mainly for its widespread availability. β-CTX has a wide range of potential applications in paediatric settings regarding either primary or secondary bone metabolism disorders [14,15,16,17].

Our observations might provide further examples for this. The patient’s initial β-CTX concentration and other metabolic parameters indicated deprived bone resorptive activity. High ALP and BAP levels can be attributed to the osteopetrotic cells and the increased synthetic activity of bones. In the case of ALP, accompanying liver damage is also a contributing factor. The transitory decrease of osteocalcin can probably be related to the damage of osteoblasts following conditioning. Donor-derived osteoclasts initiate bone remodelling during the early period of HSCT, as displayed by the rise of β-CTX levels. This primary remodelling makes the niche of the sclerotic bone marrow a more hospitable environment for haematopoietic activity and upcoming stem cells. Due to the narrowed intratrabecular space, delayed haematopoietic reconstitution is a chief concern in ARO. Therefore, to ensure engraftment, an additional stem cell boost is given [3]. Highly elevated osteoclast activity as indicated by β-CTX levels above the 97th range, the gradual correction of other biochemical parameters, and improving haematopoietic recovery demonstrated the successful remodelling following the boost. β-CTX levels were also in consonance with the improved radiographic disease phenotype.

It should also be noted that decreased bone density is a known complication of HSCT, possibly caused by the impairment of bone formation and the boost of osteoclastic activity. Polgreen et al. [9] observed the increase of several bone resorption markers, including CTX, over the first 100 days of HSCT in a population including malignant and non-malignant haematologic diseases. Thus, some aspects of the transplantation might inherently affect β-CTX kinetics. Cytotoxic agents and irradiation therapy of the conditioning, systemic glucocorticoids used to treat transplantation-related complications such as graft-versus-host disease, and the “cytokine-storm” accompanying the engraftment may be the main culprits behind this. In our case, the patient received myeloablative conditioning, a possible contributing factor for the initial β-CTX increment. There was no need for systemic glucocorticoids during the HSCT, however. Nevertheless, given the differences in the pre-and post-transplantation therapeutic course between ARO and hematologic malignancies, caution might be advised when these groups are compared.

In the case of the neurocranium, however, an irreversible bony fusion of cranial sutures (sagittal synostosis) was observed. Paradox craniosynostosis after HSCT is not unknown but is an extremely rare phenomenon remaining obscure to this day [18,19]. In this age group, the cranial sutures function as intramembranous bone growth sites with undifferentiated mesenchymal stem cells in the suture matrix and bone-forming cells by the overlap of the suture and the bone [20,21]. Craniosynostosis possibly arises with the imbalance of these differentiating, proliferating, and apoptosing cells. It seems that donor-derived osteoclasts only are not able to reverse the ossified state of the suture matrix; consequently, HSCT might not affect the osteogenic differentiation of the suture mesenchymal stem cells. Currently, the treatment of craniosynostosis is surgical, although mesenchymal stem cell therapies represent a promising new direction [22].

Traditional tools for establishing the success of HSCT and long-term remission, such as measuring donor chimerism, are constrained in the case of ARO, for they provide no information on osteoclast function. The use of β-CTX and other BTMs may be of aid in assessing osteoclast activity throughout the transplantation to monitor the effect of the stem cell boost and the resolution of ARO in a simple, fast way. Further studies in this field might have a potential for expanding the diagnostic and follow-up tools for clinicians [23,24]. Table 2 gives a summary of the few studies in this field.

Our report is to be viewed with the limitations of featuring a single case. These data must also be interpreted with caution, as the bone metabolic profile did not include relevant hormones or other commercially available BTMs, and the β-CTX values prior to the HSCT are not available.

## Figures and Tables

**Figure 1 children-10-00675-f001:**
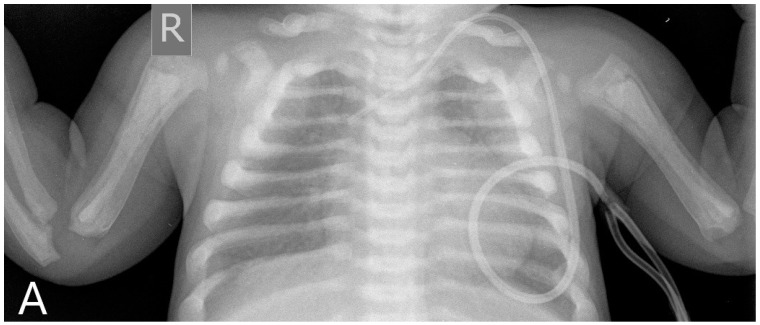
Pre-transplantation radiography (**A**) shows increased bone density, rachitic malformation in ribs, and endobone appearance in vertebrae and humeri. Sagittal T1-weighted MR image (**B**) demonstrates the sclerosis of the calvaria. Axial T2-weighted MR image (**C**) shows optic canal stenosis and optic nerve atrophy.

**Figure 2 children-10-00675-f002:**
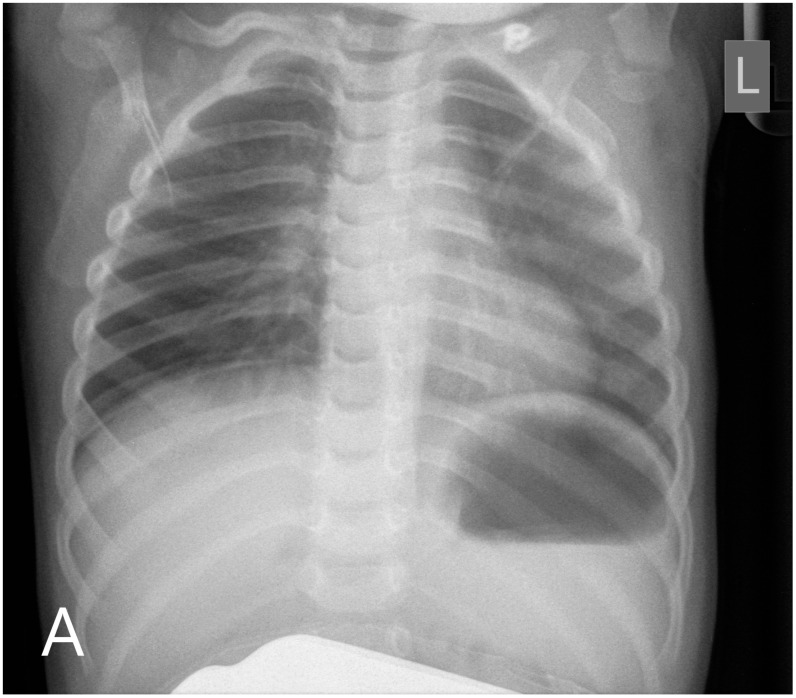
Chest radiography (**A**) shows the disappearance of skeletal malformations 6 months after the transplantation. At this time, sagittal T1-weighted MR image (**B**) described the unusual morphology of the calvaria: craniosynostosis.

**Figure 3 children-10-00675-f003:**
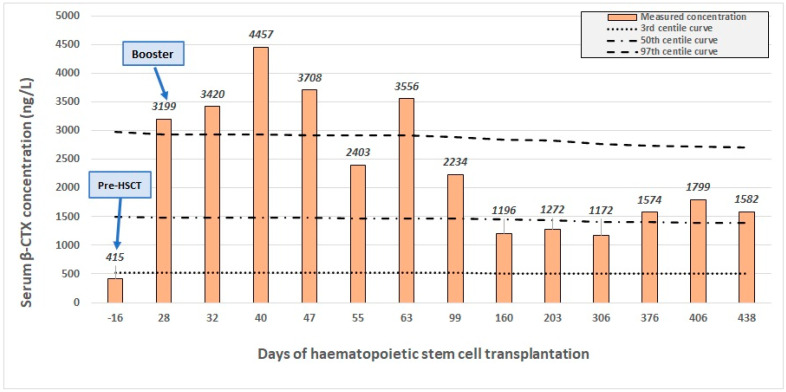
Serum concentration of β-CTX (β-C-terminal telopeptide) throughout the transplantation. The 97th, 50th, and 3rd centile curves (dotted lines) are shown besides the patient’s own concentration values (columns). Major clinical events: start of conditioning (day –6), HSCT (day 0), engraftment (day 20), booster (day 30), and osteotomy (day 364).

**Table 1 children-10-00675-t001:** Parameters of bone metabolism throughout the transplantation.

	Normal Values	−15	+30	+60	+100	+200
**Ca (mM/L)**	2.08–2.65	2.01	2.01	2.22	2.34	2.52
**Pi (mM/L)**	0.78–1.65	0.78	0.88	1.68	0.95	1.15
**ALP (U/L)**	100–350	2391	1945	608	472	361
**BAP (U/L)**	*	1864	1536	449	382	259
**OC (ng/mL)**	10–40	30.6	6.75	38.4	36	28.8

Abbreviations: Ca, calcium; Pi, phosphate; ALP, alkaline phosphatase; BAP, bone-specific alkaline phosphatase; OC, osteocalcin. * Normal range of BAP is usually determined as its percentage of total ALP, which varies between 77–88% in children.

**Table 2 children-10-00675-t002:** Studies and case reports related to the bone turnover marker profile of paediatric patients undergoing HSCT.

Authors	BTMs Evaluated		Disease Group	Number of Patients	Age/Age Range	Follow-Up	Main Findings
	Resorption Markers	Formation Markers					
Kulpiya et al. [24]	β-CTX	P1NP, OC	ARO	1	3 years	16.5 months	Markedly increased bone resorption and formation markers 43 days after HSCT. Concurrently asymptomatic hypercalcaemia developed. β-CTX levels were persistently raised during the follow-up period. Further hypercalcaemic episodes after 157 and 171 days. P1NP and OC returned to normal range.
Chen et al. [23]	TRACP5b	BAP	ARO	1	4 months	9 months	TRACP5b activity was elevated compared to control before HSCT and gradually decreased to the normal range after 3 months. The large number of dysfunctional, pre-transplant osteoclasts and their replacement by donor-derived cells may be the explanation. BAP activity appeared to be in the normal range before HSCT; a gradual decrease to low values was observed through the follow-up period. Transplantation-associated procedures may be the causative elements behind this.
Polgreen et al. [9]	CTX, NTX, DPD, PYD	-	Malignant and non-malignant haematologic diseases	26	5.2–16.6 years	6 months	Both DPD and PYD decreased over the first 30 days in the malignant group but increased afterwards and stabilised between 100 and 180 days. CTX showed an upward trend in the first 30 days and increased further to 100 days. NTX showed no significant changes. Steroid treatment resulted in a greater increase in DPD and CTX. There were no significant changes in pubertal vs. prepubertal groups. There was a greater decrease in PYD and a lower increase in CTX in males.

Abbreviations: CTX, C-terminal telopeptide; NTX, N-terminal telopeptide; P1NP, procollagen type 1 N-terminal propeptide; OC, osteocalcin; TRAC, tartarate-resistant acid phosphatase; BAP, bone-specific alkaline phosphatase; DPD, deoxypyridinoline; PYD, pyridinoline.

## Data Availability

Not applicable.

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
