# Peer review of "Bone Turnover Marker for the Evaluation of Skeletal Remodelling in Autosomal Recessive Osteopetrosis after Haematopoietic Stem Cell Transplantation: A Case Report"

_children, 2023, doi:10.3390/children10040675_

Round 1

Reviewer 1 Report

The manuscript reports the case of a 6-month-old child who received a peripheral stem cell allograft for infantile malignant osteopetrosis. The authors describe for the first time the kinetics of a bone turnover marker, β-CTX, assayed in the routine laboratory, following bone marrow allograft transplantation. 

These kinetics are particularly interesting, as β-CTX is initially below the 3rd percentile pre-transplant, then appears to rise from D28 onwards, above "norms", reflecting greatly increased bone resorption. β-CTX returns to median levels at D160 (between 5 and 6 months after allograft) and this level is stable over time until the end of the follow-up (438 days).

The authors report that despite this increased osteoclastic activity, the child presented a craniosynostosis with the need for craniotomy at 1 year after the transplant.

They also report the evolution of more usual markers such as calcemia, phosphatemia, alkaline phosphatase (and bone-specific alkaline phosphatase) and osteocalcin.

This case is therefore very interesting and deserves to be published in the journal, with some minor modifications.

1. Do the authors have several chimerism points in the follow-up of this child? If so, it would be interesting to mention them in order to correlate them with the β-CTX kinetics (adding the values on fig.3)

2. The authors should specify whether the child required corticosteroids after transplantation, which could have an impact on bone remodelling and β-CTX values. 

3. As the child had an unfavourable evolution of the skull bones, and a  blindness, it would be interesting to know what was his auditory evolution, in order to confirm that the bone remodelling observed on the radiographs had a positive impact on the sensory evolution.

4. In figure 3, do the authors have several β-CTX values before transplantation? If so, it would be interesting to mention several. (Alternatively, if a value before D28 is also available, it would be useful to show it).

5. In figure 3, the norms (97th, 50th and 3rd percentiles) are taken from one reference: this should be cited, and added to the citations at the end of the manuscript (reference 8 omitted).

6. In the discussion, it would be interesting to briefly discuss other bone turnover markers already reported in the literature in osteopetrosis, and their difference in interpretation from β-CTX: e.g. Tartrate resistant acid phosphatase 5b TRACP5b (Chen et al, J Pediatr Hematol Oncol 2004 mentioned in the references) Line 195 "promising BTMs" should be changed as the references date from 2004 and 2012, the term promising seems inappropriate.

7. In the discussion, it would also be interesting, if already reported, to give a frequency and pathophysiological explanation for craniosynostoses that continue to evolve despite transplantation.

8. In the discussion, the authors could mention very briefly that the initial elevation of β-CTX above norms could be related to the increased osteoclastic activity described in allogeneic hematopoietic transplantation for other conditions than osteopetrosis between D30 and D100 (therefore not disease dependent Polgreen et al, Pediatr Transplant 2012).

Reviewer 2 Report

This study presents a case report of a patient diagnosed with a rare bone disorder, autosomal recessive osteopetrosis (ARO), who received an allogeneic hematopoietic stem cell transplant (allo-HSCT). Horváth et al. observed a significant increase in the levels of β-carboxy-terminal telopeptide (β-CTX) in the patient's serum after allo-HSCT. This finding suggests that β-CTX may serve as a potential bone turnover marker for ARO patients undergoing allo-HSCT. Although the findings are limited by the single case report, they provide valuable insight into bone remodeling evaluation for ARO patients undergoing allo-HSCT.

Several recommendations for future studies are suggested. 

1. An abstract should be included to provide a concise summary of the study's objectives, methods, and key findings. 

2. While the authors measured β-CTX in the patient's serum, it is recommended that β-CTX levels in the patient's urine be evaluated as well. 

3. The measurement of other bone turnover markers such as bone-specific alkaline phosphatase (BALP), osteocalcin, and procollagen type 1 N-terminal propeptide (P1NP) should also be included for a more comprehensive evaluation of bone remodeling.

Round 2

Reviewer 2 Report

No further comment